# Antimicrobial Resistance Genes and Diversity of Clones among ESBL- and Acquired AmpC-Producing *Escherichia coli* Isolated from Fecal Samples of Healthy and Sick Cats in Portugal

**DOI:** 10.3390/antibiotics10030262

**Published:** 2021-03-05

**Authors:** Isabel Carvalho, Nadia Safia Chenouf, Rita Cunha, Carla Martins, Paulo Pimenta, Ana Raquel Pereira, Sandra Martínez-Álvarez, Sónia Ramos, Vanessa Silva, Gilberto Igrejas, Carmen Torres, Patrícia Poeta

**Affiliations:** 1Microbiology and Antibiotic Resistance Team (MicroART), Department of Veterinary Sciences, University of Trás-os-Montes and Alto Douro, 5000-801 Vila Real, Portugal; isabelbarrosocarvalho@utad.pt (I.C.); vanessasilva@utad.pt (V.S.); 2Department of Genetics and Biotechnology, UTAD, 5000-801 Vila Real, Portugal; gigrejas@utad.pt; 3Functional Genomics and Proteomics Unit, UTAD, 5000-801 Vila Real, Portugal; 4Laboratory Associated for Green Chemistry (LAQV-REQUIMTE), New University of Lisbon, 2829-516 Monte da Caparica, Portugal; 5Area Biochemistry and Molecular Biology, University of La Rioja, 26006 Logroño, Spain; chenoufns@gmail.com (N.S.C.); sandra.martinezal@alum.unirioja.es (S.M.-Á.); carmen.torres@unirioja.es (C.T.); 6Laboratory of Exploration and Valuation of the Steppe Ecosystem, University of Djelfa, Djelfa 17000, Algeria; 7Hospital Veterinário de São Bento, 1200-822 Lisboa, Portugal; ritalaurentinocunha@gmail.com; 8Clínica Veterinária do Vouga, 3740-253 Sever do Vouga, Portugal; clinica.vouga@gmail.com; 9Hospital Veterinário de Trás-os-Montes, 5000-056 Vila Real, Portugal; paulo.pimenta@onevetgroup.pt; 10Centro Veterinário de Macedo de Cavaleiros, 5340 Bragança, Portugal; vetcenterbrg@gmail.com; 11VetRedondo, Consultório Veterinário de Monte Redondo Unipessoal Lda, Monte Redondo, 2425-618 Leiria, Portugal; soniacatarinaramos@gmail.com

**Keywords:** antimicrobial resistance, *E. coli*, cats, public health, extended spectrum beta-lactamase (ESBL), AmpC beta-lactamase, CTX-M, CMY-2

## Abstract

The aim of the study was to analyze the mechanisms of resistance in extended-spectrum beta-lactamase (ESBL)- and acquired AmpC (qAmpC)-producing *Escherichia coli* isolates from healthy and sick cats in Portugal. A total of 141 rectal swabs recovered from 98 sick and 43 healthy cats were processed for cefotaxime-resistant (CTX^R^) *E. coli* recovery (in MacConkey agar supplemented with 2 µg/mL cefotaxime). The matrix-assisted laser desorption/ionization time-of-flight mass spectrometry (MALDI-TOF-MS) method was used for *E. coli* identification and antimicrobial susceptibility was performed by a disk diffusion test. The presence of resistance/virulence genes was tested by PCR sequencing. The phylogenetic typing and multilocus sequence typing (MLST) were determined by specific PCR sequencing. CTX^R^
*E. coli* isolates were detected in seven sick and six healthy cats (7.1% and 13.9%, respectively). Based on the synergy tests, 11 of 13 CTX^R^
*E. coli* isolates (one/sample) were ESBL-producers (ESBL total rate: 7.8%) carrying the following ESBL genes: *bla*_CTX-M-1_ (*n* = 3), *bla*_CTX-M-15_ (*n* = 3), *bla*_CTX-M-55_ (*n* = 2), *bla*_CTX-M-27_ (*n* = 2) and *bla*_CTX-M-9_ (*n* = 1). Six different sequence types were identified among ESBL-producers (sequence type/associated ESBLs): ST847/CTX-M-9, CTX-M-27, CTX-M-1; ST10/CTX-M-15, CTX-M-27; ST6448/CTX-M-15, CTX-M-55; ST429/CTX-M-15; ST101/CTX-M-1 and ST40/CTX-M-1. Three of the CTX^R^ isolates were CMY-2-producers (qAmpC rate: 2.1%); two of them were ESBL-positive and one ESBL-negative. These isolates were typed as ST429 and ST6448 and were obtained in healthy or sick cats. The phylogenetic groups A/B1/D/clade 1 were detected among ESBL- and qAmpC-producing isolates. Cats are carriers of qAmpC (CMY-2)- and ESBL-producing *E. coli* isolates (mostly of variants of CTX-M group 1) of diverse clonal lineages, which might represent a public health problem due to the proximity of cats with humans regarding a One Health perspective.

## 1. Introduction

The emergence and fast dissemination of antimicrobial-resistant bacteria (AMR) continue to be a public health concern in both medicine and agriculture [1]. It was estimated that the global consumption of antimicrobial agents in animal food production will increase by 67% between 2010 and 2030 [2]. Accordingly, this global rapid increase of AMR has been mostly attributed to the overuse and misuse of antibiotics in human and veterinary medicine [3,4]. In the veterinary environment, antimicrobial use is discussed as one of the main drivers of antimicrobial resistance development [5,6].

*Escherichia coli*, a Gram-negative bacterium that normally habits in the gastrointestinal tract of healthy humans and warm-blooded animals, is also an important opportunistic pathogen [6,7]. This commensal microorganism is known to be an important indicator of the antibiotic resistance evolution as well as an eventual reservoir of virulence genes in different ecosystems [8,9].

Urinary tract infections (UTIs) are one of the common health problems among companion animals in European countries. Gram-negative bacteria, mainly *E. coli* strains, are responsible for 75% of the cases [6,10]. Treatment of UTIs caused by *E. coli* strains is becoming difficult due to the antibiotic resistance phenomenon. *E. coli* can act as a reservoir of antimicrobial-resistant genes that can be transmitted to other pathogenic bacteria. Multidrug-resistant (MDR) microorganisms are transmitted among pets, owners and veterinary staff, which leads to their spread within the community [11,12,13].

Extended-spectrum β-lactamase (ESBL), acquired AmpC β-lactamase (qAmpC) and carbapenemase production in *Enterobacteriaceae* are three of the major concerns due to their rapid emergence in humans and animals during the last years [9,13,14,15].

*Escherichia coli* isolates harbor intrinsic and low-expressed chromosomic AmpC β-lactamases (cAmpC) that do not confer resistance to β-lactams; nevertheless, when overexpressed (generally by mutations at either the promoter or the attenuator of the structural gene), they produce a resistance to amoxicillin-clavulanic acid, cefoxitin and third generation cephalosporins [16]. *E. coli* isolates can also acquire different AmpC β-lactamases (qAmpC), also conferring an AmpC phenotype. During the last years, different qAmpC types have been reported (as CMY, DHA, MOX, FOX and ACC) or the qAmpC β-lactamases family (EBC), being genes that are mostly plasmid located. The first plasmid-mediated qAmpC, named CMY-1, was described in South Korea (1989). Recent studies have described an increase in specific qAmpC enzymes among hospital or community infections caused by *Enterobacteriaceae* [17,18]. The DHA-1 and CMY-2 variants are the most common qAmpC types among *Enterobacteriaceae* implicated in human infections in Portugal [18]. During the last years, the CMY-2 enzyme has been increasingly reported in humans and animals, frequently associated with lineages ST429, ST354 or ST57, among others [17,19,20,21]. According to recent studies in Portugal, qAmpC-producing *E. coli* was reported (CMY-2) among sick pets, mostly associated with ST648 [6], and among healthy humans, associated with lineages ST4953 and ST665 [22]. Moreover, CMY-2 was also reported among Portuguese clinical *E. coli* isolates associated with a diversity of lineages [6,18].

The number of reports concerning ESBLs from *E. coli* isolates in companion animals has also increased [23,24]. Moreover, CTX-M type enzymes have formed a rapidly growing family of ESBLs in bacteria from human infections. According to a review done by Dandachi et al. [25], the CTX-M group 1 seems to prevail in animals in the Mediterranean Basin followed by SHV-12 and CTX-M group 9. The pandemic-specific ST131 clone with multiresistance and a high virulence potential is largely associated with the global increase of these ESBL-producers.

In addition, shiga toxin (stx)-producing *Escherichia coli* (STEC) strains have emerged as an important cause of serious human gastrointestinal disease during the last years. Pets can also be reservoirs of STEC and enteropathogenic *E. coli* (EPEC) strains. Recent studies have reported its presence among *E. coli* isolated from wildlife animals [14,26].

Companion animals (including cats) may play an important role in the spread of resistant bacteria due to their frequent and close contact with humans. Therefore, the risk of animal to human transference of such bacteria is a significant concern [3]. Regarding the global situation of antimicrobial resistance, many studies have been performed and current data are available particularly in human settings [6,9]. Likewise, other authors either from Europe [27,28,29] or from other continents [30,31,32,33] have focused on the study of *E. coli* isolates from pets. In Portugal, studies about AMR with a detection of ESBL and/or qAmpC in *E. coli* isolates recovered from humans [18,34,35] and healthy or sick pets [36,37,38] were communicated. Despite the scarcity of data about healthy and sick cats, all cited surveys showing a worrying picture.

To our knowledge, this study is the first report that combines AMR among *E. coli* isolates from both healthy and sick cats in Portugal. It aimed to analyze the occurrence and molecular epidemiology of qAmpC- and ESBL-producing *E. coli* isolates originating from healthy cats (kennel and house cats) and sick cats (of seven different hospitals) in the Portuguese territory.

## 2. Results

### 2.1. Cefotaxime-Resistant (CTX^R^) E. coli Isolates and Antimicrobial Susceptibility Testing

Of the 141 fecal samples analyzed (98 from sick cats and 43 from healthy cats), CTX^R^
*E. coli* isolates were detected in 13 (seven from sick and six from healthy cats; rates of 7.1% and 13.9%, respectively) and one isolate per positive sample was further characterized (corresponding to 12 mixed breed cats and one *Siamês*). Regarding the healthy cats, five of the positive samples came from healthy cats from their owners and the other sample from a cattery located in Vila Real (Table 1). Regarding the sick animals, CTX^R^
*E. coli* isolates were recovered from samples of five different veterinary clinics/hospitals (Figure 1) distributed as follows: Vila Real (two isolates from Veterinary Clinic Quinchosos and one isolate from Transmonvete), Bragança (two isolates from Veterinary Clinic Macedo de Cavaleiros) and Leiria (two isolates from Veterinary Clinic Guia) (Table 1).

The 13 CTX^R^
*E. coli* isolates showed high rates of resistance for cefoxitin (61.5%), amoxicillin + clavulanic acid (76.9%), tetracycline (53.8%), ciprofloxacin (38.5%) and trimethoprim + sulfamethoxazole (30.8%) (Table 1). Resistance to chloramphenicol, gentamicin and streptomycin were also observed in three isolates in each one (23%). Consequently, most of the isolates showed a multidrug-resistance (MDR) phenotype (92.3%) including resistance to at least three classes of antimicrobial agents.

### 2.2. Genes Encoding ESBLs and qAmpC Enzymes as Well as Other Resistance/Virulence Genes

Based on the synergy tests, 11 *E. coli* isolates were found to be ESBL-producers (7.8% of total samples; 11.6% of healthy cats and 8.1% of sick cats). Five different *bla*_CTX-M_ variants were detected among our ESBL-producing *E. coli* (number of isolates): *bla*_CTX-M-1_ (3), *bla*_CTX-M-15_ (3), *bla*_CTX-M-55_ (2), *bla*_CTX-M-27_ (2) and *bla*_CTX-M-9_ (1) (Table 1). Furthermore, the *bla*_TEM_ gene was detected in eight of the 11 ESBL-positive *E. coli* isolates (72.7%) in association with the *bla*_CTX-M_ gene. One isolate carried the *bla*_SHV-28_ gene together with *bla*_CTX-M-27_ and *bla*_TEM_ genes.

Eight of the 13 CTX^R^ isolates showed a resistance to cefoxitin and amoxicillin clavulanic acid, a phenotype compatible with a qAmpC. After testing the presence of qAmpC genes by PCR, three isolates gave a positive result for the *bla*_CMY-2_ gene (two ESBL-positive and one ESBL-negative); these isolates were obtained from two healthy cats and one sick cat. Taken together, qAmpC-carrying isolates were detected in three of the 141 animals tested (2.1%) and in two cases associated with ESBLs.

Considering all of the 13 CTX^R^ isolates, the *tet*A gene was detected among seven tetracycline-resistant isolates and the combination of both *tet*A and *tet*B genes was detected in one additional isolate (Table 1). The integrase of class 1 integrons (*int*1) was revealed among four SXT-resistant isolates and they carried the *sul1* and/or *sul2* genes. Three chloramphenicol-resistant isolates were detected but they were negative for *cml*A, *cat*A and *floR* genes. Two ciprofloxacin-resistant isolates carried the *aac(6**’)-Ib-cr* gene and one amikacin-resistant isolate carried the *armA* gene. None of the CTX^R^ isolates contained the *mcr*_1,4_ resistance genes nor the *sxt-1*, *sxt-2* or *eae* virulence genes.

### 2.3. Phylogenetic Groups and MLST Typing

The 13 CTX^R^
*E. coli* isolates were ascribed to phylogroups B1 (*n* = 4), A (*n* = 3), D (*n* = 2) and clade 1 (*n* = 1); three additional isolates were not conclusive regarding the phylogroup guidelines according with Clermont et al. [39]. MLST was performed in the twelve ESBL- and qAmpC-producing *E. coli* isolates and the following sequence types were obtained (sequence type/associated ESBL types (phylogroups)): ST847/CTX-M-9 or CTX-M-27 or CTX-M-1 (B1, one isolate); ST10/CTX-M-15 or CTX-M-27 (A); ST6448/CTX-M-15 or CTX-M-55 (B1); ST429/CTX-M-15 (clade 1); ST101/CTX-M-1 (D); ST40/CTX-M-1 (B1) and ST429/CMY-2 (D) (Table 1). A diversity of CTX-M variants was detected among healthy (CTX-M-1, -9, -15 and-27) and sick cats (CTX-M-1, -15 and -55).

## 3. Discussion

Worldwide, it is known that antimicrobial-resistant bacteria have hugely disseminated in both clinical and non-clinical settings. In our study, 141 fecal samples were studied from healthy and sick cats in Portugal to estimate the occurrence of ESBL- and qAmpC-producing *E. coli* isolates. Almost 8% of the total fecal samples analyzed contained ESBL-producers and 2.1% of samples were confirmed as qAmpC-producers (carrying the *bla*_CMY-2_ gene). Two isolates co-produced ESBLs and qAmpC enzymes. We could not detect the mechanism of β-lactam resistance in one CTX^R^ ESBL-negative *E. coli* isolate with an AmpC phenotype; however, it cannot be discarded the presence of other non-tested acquired *amp*C genes or the hyperproduction of the chromosomal *amp*C gene. One limitation of this study is the selection of only one CTX^R^ isolate/positive sample; with more variability of ESBL or qAmpC-producers we could have detected if more CTX^R^ isolates were selected in each positive sample.

The detection of CMY-2-producing *E. coli* isolates was previously reported in healthy and sick dogs in Italy [29,40], healthy dogs in Mexico [21] and sick dogs and cats in Portugal [6] and the United States [41] among others. Moreover, the CMY-2 enzyme was recently reported among healthy humans in Portugal (0.48%) [22] as well as in human clinical settings (1.04%) [6] with a similar prevalence as the one detected in our study (2.1%). The One Health perspective can be considered because of possible horizontal transmission among pets and humans. Freitas et al. [34] detected the CMY-2 encoding gene among 6% (3/50) of qAmpC-producers in different Portuguese hospitals.

Our findings confirm that ESBLs in companion animals have increased rapidly as mentioned by other authors [24,42,43]. In Brazil, a rate of 24.8% was found in healthy stray cats [32]. However, no ESBL-positive isolates were identified in other studies done in healthy cats from Portugal [38] or in sick cats in Turkey [10]. Furthermore, no CTX^R^ was found in *E. coli* isolates from cats in Oporto (Portugal) [44]. In the Netherlands, 25% of cats with diarrhea carried ESBL-producing isolates and none of the healthy cats’ isolates (0%) was positive for ESBLs [43]. Furthermore, Piccolo et al. [40] recently found 26.2% of ESBL-producing *E. coli* among diseased cats in Italy. Moreover, a high rate of ESBL-producing *E. coli* isolates was reported among clinical samples from pets in Singapore (40%) [31]. Curiously, in our study, the ESBL prevalence was higher in healthy cats (11.6.%) than in sick animals (8.1%); although the reason for this difference is unknown, we cannot discard the influence of the different number of animals included in each group. In this respect, Hordijk et al. reported that 25% of sick cats were ESBL-producers but no ESBL production was detected among healthy cats in the Netherlands [45]. Different studies have been performed in Portugal regarding ESBL-producing *E. coli* isolates recovered from sick and healthy dogs and cats [6,38]. Nevertheless, to our knowledge, this is the first report of the characterization of qAmpC- and ESBL-producing *E. coli* isolates from healthy and sick cats simultaneously in the same work and country. Moreover, our results about the MDR phenotype join information with other studies already performed in isolates of companion animals from different origins and countries [24,30,40].

In the present report, the high abundance of the *bla*_CTX−M−1_ and *bla*_CTX−M−15_ variants among the ESBL-positive isolates (6/11) (Table 1) was consistent with studies based on isolates from humans and farm animals in which *bla*_CTX−M−1_ was either the most or second most predominant ESBL gene. It is important to note that CTX-M-15 has emerged as the predominant ESBL type in *E. coli* isolates in healthy and sick humans [9,46]. Our results are also in accordance with the data reported by Day et al. [15] in which the prevalence of *bla*_CTX-M-1_ and *bla*_CTX-M-15_ in human isolates from the Netherlands and Germany was high. Furthermore, Hordijk et al. [43] indicated that diarrheic cats were positive for *bla*_CTX−M−1_ and *bla*_CTX−M−15_. The first report of CTX-M-1 in Portugal was performed in one healthy dog [38]. Carattoli et al. [29] also reported the *bla*_CX-M-1_ gene in two healthy cats in Italy. Furthermore, Marques et al. [6] observed that five out of 220 cats with a UTI carried *E. coli* isolates with CTX-M-15, CTX-M-1, CTX-M-32 or CTX-M-9 variants.

The phylogenetic group B_1_ represented the most prevalent group followed by A (*n* = 3) and D (*n* = 2) among the CTX^R^ isolates of this study. The phylogenetic group B2 was nevertheless the most frequently detected among *E. coli* isolates implicated in urinary tract infections of cats and dogs in a previous study [10].

In the last decade, *E. coli* of clonal group ST131 has emerged as an important clinical health concern causing multidrug-resistant (MDR) infections worldwide in animals and humans [9,47]. However, this epidemic clone was not found among these cats’ fecal samples. Concerning the sequence types found in this study, ST847 (*n* = 3), ST6448 (*n* = 3), ST10 (*n* = 2), ST101, ST429 and ST40 were detected in *bla*_CTX-M-1,9,27_, *bla*_CTX-M-15,55_/*bla*_CMY-2_, *bla*_CTX-M-15,27_, *bla*_CTX-M-1_, *bla*_CTX-M-15_*/bla*_CMY-2_ and *bla*_CTX-M-1_-containing strains, respectively (Table 1). A previous report has indicated the detection of ST847 among avian ESBL-*E. coli* isolates in Germany [48]. The same authors verified that the most common multilocus sequence types (STs) in ESBL-producers, for which whole genome sequencing was performed, were ST131 (25.6%) followed by ST10. Day et al. [15] reported also the presence of ST10 from human isolates in Germany and the Netherlands. To our knowledge, the ST101 is an international sequence type frequently detected in pigs [49] and broilers [50] as well as in clinical settings; this clone can be considered to be a candidate for the zoonotic transmission to the human population.

In order to compare the data of healthy and sick cats, the healthy cats were frequently colonized by ESBL-producing *E. coli* (different variants of CTX-M enzymes; namely, CTX-M-15,27,9) and qAmpC production (associated with CMY-2) with a predominance of ST429 and ST6448. On the other hand, sick cats were associated with CTX-M-1,15,55 and with a predominance of ST6448. In all cases, the heterogenicity of phylogroups was observed.

The ST6448 lineage was detected in three of our isolates carrying the genes encoding CTX-M-15 or CTX-M-55 (associated with CMY-2 in one isolate); this lineage has also been found in previous studies of our team in two CTX-M-15-producing isolates of sick dogs in Portugal [51] and in one CTX-M-55-producing isolate of a vulture from the Canary Islands [52]. To our knowledge, there is only one previous report related to the detection of this clone in humans, performed in healthy children in Sweden [53]. Furthermore, this clone was also detected in *E. coli* isolates from broilers in Algeria, associated with the B1 phylogroup [54].

Regarding the ST40 clone, a relation between diarrheal disease and the presence of this lineage (containing specific virulence genes) was recently found among clinical isolates from patients in the United Kingdom [55]. Furthermore, the ST429 lineage was found in our study in CMY-2- and CTX-M-15-producing isolates of healthy cats. The ST429 lineage was previously reported among CMY-2-producing isolates of a poultry origin in Spain [50], Germany [20] and, more recently, in the Czech Republic [56], mostly implicated in avian diseases [56]. This lineage was not detected among the CMY-2-producing *E. coli* isolates detected in humans in the north of Portugal [18,22,34]. However, CMY-2 was identified among *Enterobacteriaceae* in a Portuguese hospital [57].

## 4. Materials and Methods

### 4.1. Animals and Sampling

A total of 141 fecal samples were collected using standardized procedures from 98 hospitalized cats (69.5%) recovered from seven different veterinary clinic/hospitals and from 43 healthy cats (30.5%), from three local kennels and 40 local houses between January and April 2017 (one sample per animal). Veterinary clinics/hospitals were located in the following Portuguese territories: Braganza (A), Vila Real (B, C, D, E), Aveiro (F), Leiria (G) and Lisbon (H) (Figure 1). The samples of sick cats were collected from those animals that attended the hospital and stayed hospitalized; in the case of healthy cats, the fecal samples were collected after obtaining the owner’s permission or in collaboration with a local kennel. The samples were then dispatched immediately to the Microbiology Laboratory of the University of Trás-os-Montes and Alto-Douro (UTAD) located in Vila Real (Portugal) for processing.

### 4.2. Escherichia Coli Isolation

From each fecal sample, a small portion of 2 g was taken and diluted in Brain Heart Infusion (BHI) broth and incubated in an aerobic condition for 24 h at 37 °C. They were then inoculated with swabs onto MacConkey agar plates supplemented with 2 μg/mL of cefotaxime (MC + CTX). The plates were then incubated at 37 °C during 24 h and one colony per sample with the morphological aspect of *E. coli* was selected and identified by conventional biochemical tests (IMViC: Indol, Methyl-red, Voges–Proskauer and Citrate).

The matrix-assisted laser desorption/ionization time-of-flight mass spectrometry method (MALDI-TOF MS, Bruker) was applied in this study to confirm the bacterial species identification and *E. coli* isolates were kept at −80 °C for further analysis.

### 4.3. Susceptibility Testing

Antimicrobial susceptibility testing was performed using the Kirby–Bauer disk diffusion method on Mueller–Hinton agar according to the Clinical Laboratory Standards Institute guidelines [58]. The susceptibility of isolates was tested for the following antibiotics (μg/disk): ampicillin (10), amoxicillin + clavulanic acid (20/10), cefoxitin (30), cefotaxime (30), ceftazidime (30), aztreonam (30), imipenem (10), gentamicin (10), amikacin (30), streptomycin (10), nalidixic acid (30), ciprofloxacin (5), tetracycline (30), trimethoprim-sulfamethoxazole (1.25 ± 23.75) and chloramphenicol (30). The plates were incubated aerobically for 24 h at 37 °C.

In addition, the screening of phenotypic ESBL production was carried out in the CTX^R^ isolates by the double disk synergy test using cefotaxime, ceftazidime and amoxicillin/clavulanic acid discs [58]. A presumptive qAmpC phenotype was assigned to the CTX^R^
*E. coli* isolates that showed resistance to cefoxitin (FOX^R^) and intermediate susceptibility or resistance to amoxicillin-clavulanic acid (AMC^I/R^) [34,59]; the presence of qAmpC resistance genes was tested in those isolates.

### 4.4. Antibiotic Resistance Genes

Genomic DNA was obtained from *E. coli* isolates using the boiling lysis method [60]. The presence of antibiotic resistance genes was investigated by PCR and subsequent sequencing of the obtained amplicons (when required). The presence of β-lactamase genes was analyzed in the CTX^R^ isolates (*bla*_TEM_, *bla*_SHV_, *bla*_OXA_ and *bla*_CTX-M_ of different groups) and the genes *bla*_CMY_, *bla*_DHA_, *bla*_FOX_, bla_MOX_ and *bla*_ACC_ were also tested in the isolates with a presumptive qAmpC phenotype; all of these isolates were also tested for the *mcr*-1 and *mcr-*4 genes associated with colistin resistance [61,62,63,64]. Resistance phenotypes and isolates were also checked (by PCR sequencing when required) for gene encoding resistance to tetracycline (*tet*(A) and *tet*(B)), sulfonamides (*sul1* and *sul2*), chloramphenicol (*cat*A, *cml*A and *flor*R), ciprofloxacin (*aac*(6´)-*Ib-cr*) and amikacin (*aph*(3´)-VI and *armA*) [62,63,64,65,66]. The analysis of DNA sequences was performed with the BLAST program available at the National Center for Biotechnology Information.

Moreover, the presence of the *int*1 gene, encoding the integrase of class 1 integrons, was also analyzed [67] as well as the presence of *eae* and *sxt*_1,2_ genes (encoding the intimin and the shiga toxin, respectively) [19]. Positive and negative controls of the collection of the University of La Rioja (Spain) were included in all assays.

### 4.5. Molecular Typing of E. coli Isolates

The multilocus sequence typing (MLST) with seven housekeeping genes (*fum*C, *adk, pur*A*, icd, rec*A*, mdh* and *gyr*B) was carried out according to the protocol on the PubMLST (Public databases for molecular typing and microbial genome diversity) website (https://pubmlst.org/escherichia/) (access date on 1 January 2021). The allele combination was determined after sequencing the seven genes amplified by PCR and the sequence type (ST) was identified. A phylogenetic classification of the isolates was also performed according to the existence of *arp*A, *chu*A, *yja*A and TSPE4.C2 genes as previously reported by Clermont et al. combined with the presence of the E- or C-specific primers [39]. In this line, the *E. coli* isolates were assigned to one of the eight phylogenetic groups (A, B1, B2, C, D, E, F and clade 1) by multiplex PCR.

## 5. Conclusions

The present survey showed an increased risk of AMR in fecal *E. coli* strains from cats, which is consistent with the results of several previous studies in different animal species.

Although the prevalence of ESBL-producing *E. coli* in cats was higher than the corresponding qAmpC production (7.8% versus 2.1%), cats are frequent carriers of acquired mechanisms associated with CTX^R^ in *E. coli* isolates, harboring mostly the genes encoding CTX-M-1, CTX-M-15 and CTX-M-55 enzymes as well as the CMY-2 gene. Regarding the One Health approach, these resistance genes can be transferred among cats to humans by the transference of resistant bacteria or resistance genes. It is important to note that a high diversity of genetic lineages/sequence types was found in this research. To our knowledge, this is the first time that ESBL-encoding genes have been studied simultaneously in healthy and sick cats in Portugal as well as one of the few providing qAmpC-producing *Enterobacteriaceae* occurrence in healthy pets in this country. More studies should be carried out in the future to track the evolution of this type of β-lactamase in different environments.

## Figures and Tables

**Figure 1 antibiotics-10-00262-f001:**
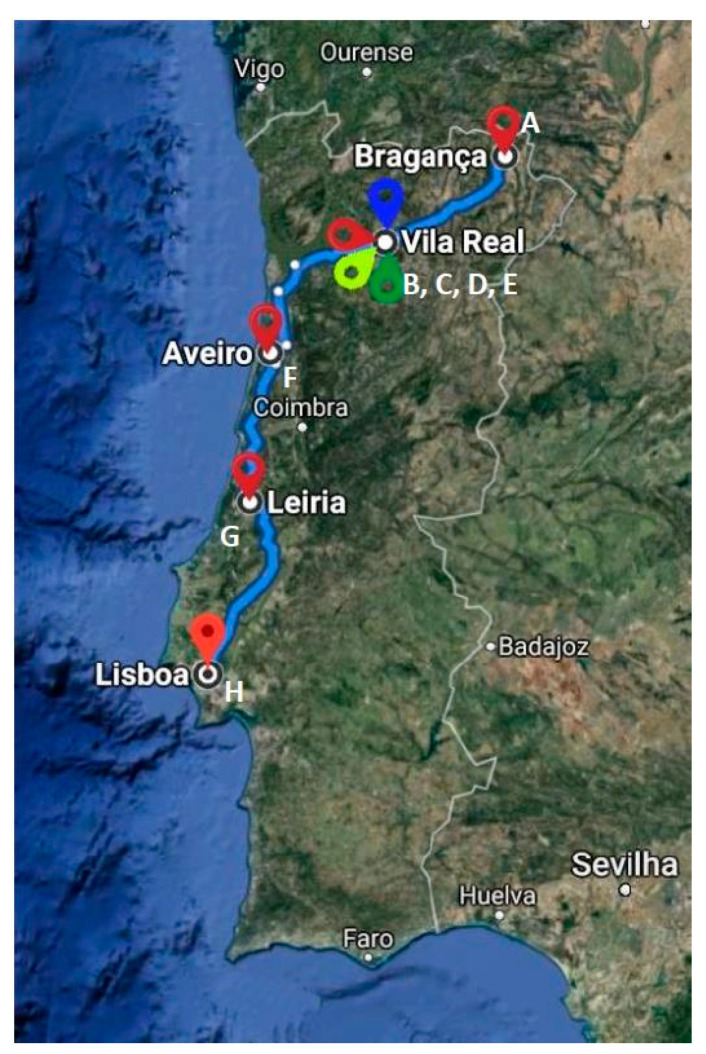
Geographic location of the different areas where the fecal samples from cats were collected in Portugal. A—Veterinary clinic from Macedo de Cavaleiros (Bragança); B—Kennel (Vila Real); C—Veterinary clinic from Quinchosos (Vila Real); D—Transmonvete (Vila Real), E—Veterinary hospital from Trás os Montes HVTM (Vila Real); F—Veterinary clinic from Vouga (Aveiro); G—Veterinary clinic from Guia (Leiria); H—Veterinary hospital from São Bento (Lisboa).

**Table 1 antibiotics-10-00262-t001:** Phenotypic and molecular features of the CTX^R^
*E. coli* isolates recovered from healthy and sick cats in Portugal.

Isolate Number	Origin ^a^	ESBL ^b^	Phenotype of Antimicrobial Resistance ^c^	Beta-Lactamases	Other Resistance Genes	PG ^d^	MLST ^e^
C10285	HC	P	AMP, AUG, FOX, CTX, CAZ, CN, S, CIP, TET	CTX-M-15, CMY-2	*aac(6’)-Ib-cr*, *tet*A	Clade 1	ST429
C10289	VCQ	P	AMP, AUG^I^, FOX, CTX, CAZ, ATM, NA, CIP	CTX-M-15, TEM		A	ST10
C10290	VCQ	P	AMP, AUG, FOX, CTX, CAZ, ATM, CHL, NA, CIP, SXT, TET	CTX-M-15, TEM	*int*1, *tet*A, *tet*B, *sul1*, *sul2*	NTYP ^d^	ST6448
C10291	VCT	P	AMP, AUG, FOX, CTX, CAZ, ATM, CHL, NA, CIP, SXT, TET	CTX-M-55, TEM, CMY-2	*int*1, *tet*A, *sul2*, *aac(6**’**)-Ib-cr*	B1	ST6448
C10295	VCM	P	AMP, AUG, CTX, CAZ, ATM, CHL, NA, CIP, SXT, TET	CTX-M-55	*int*1, *tet*A, *sul1*, *sul2*	B1	ST6448
C10282	HC	P	AMP, CTX	CTX-M-27, TEM-1		B1	ST847
C10284	HC	P	AMP, AUG, FOX, CTX, CAZ	CTX-M-27, TEM, SHV-28		A	ST10
C10283	Kennel	P	AMP, AUG, CTX, CAZ, TET	CTX-M-1, TEM	*tet*A	NTYP ^d^	ST847
C10299	VCG	P	AMP, CTX, CAZ	CTX-M-1		B1	ST40
C10293	VCM	P	AMP, AUG, FOX, CTX, TET	CTX-M-1, TEM	*tet*A	D	ST101
C10281	HC	P	AMP, AUG, CTX	CTX-M-9, TEM		NTYP ^d^	ST847
C10286	HC	N	AMP, AUG, FOX, CTX, CAZ, CN, S, TET	CMY-2	*tet*A	D	ST429
C10303	VCG	N	AMP, AUG, FOX, CTX, CAZ, NA, CIP, TOB, AK, CN, SXT, S	ND ^f^	*int*1, *arm*A, *sul1*	A	NT ^g^

^a^ HC—Healthy cats from their owners; VCQ—Veterinary clinic Quinchosos (Vila Real, Portugal); VCT—Veterinary clinic Transmonvete (Vila Real, Portugal); VCM—Veterinary clinic Macedo de Cavaleiros (Bragança, Portugal); VCG—Veterinary clinic Guia (Leiria, Portugal); VCV—Veterinary clinic Vouga (Sever do Vouga, Portugal); ^b^ P—Positive; N—Negative; ^c^ AMP, ampicillin; AUG, amoxicillin-clavulanic acid; FOX, cefoxitin; CTX, cefotaxime; CAZ, ceftazidime; ATM, aztreonam; CHL, chloramphenicol; NA, nalidixic acid; CIP, ciprofloxacin; TOB, tobramycin; AK, amikacin; CN, gentamicin; SXT, trimethoprim-sulfamethoxazole; S, streptomycin; TET, tetracycline; IMP, imipenem. I in superscript: intermediate resistance; ^d^ PG—Phylogenetic group according to the strategy of Clermont et al. [39]. Three isolates were non-typeable (NTYP) with this strategy but, as recommended, the MLST was performed; ^e^ MLST—multilocus sequence typing; ^f^ ND—non detected; ^g^ NT—non tested.

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
