# Peer review of "Antimicrobial Resistance Genes and Diversity of Clones among ESBL- and Acquired AmpC-Producing Escherichia coli Isolated from Fecal Samples of Healthy and Sick Cats in Portugal"

_antibiotics, 2021, doi:10.3390/antibiotics10030262_

Round 1

Reviewer 1 Report

General comments: In this work, the authors investigated the antibiotic resistance profile and the genetic mechanisms in ESBL-producing E. coli isolates from healthy cats and sick cats in Portugal. The research theme is interesting, and the manuscript is generally well-written. In the reviewer’s point of view, however, the research is too superficial, and no interesting results were found. 1. The isolate number is limited, and the readers cannot find the difference/relationship between the E. coli isolates from healthy cats and sick cats. 2. The conclusion section is too long, and please delete the first paragraph of the conclusion section. 3. Too many references, please delete some of them.

Author Response

General comments: In this work, the authors investigated the antibiotic resistance profile and the genetic mechanisms in ESBL-producing E. coli isolates from healthy cats and sick cats in Portugal. The research theme is interesting, and the manuscript is generally well-written. In the reviewer’s point of view, however, the research is too superficial, and no interesting results were found. 1. The isolate number is limited, and the readers cannot find the difference/relationship between the E. coli isolates from healthy cats and sick cats. 2. The conclusion section is too long, and please delete the first paragraph of the conclusion section. 3. Too many references, please delete some of them.

Answer: Thank you for the comments and suggestions. In the revised version, we have included a clear comparison of data obtained in healthy and sick cats in relation with ESBL (11.6% and 8.1%, respectively) and qAmpC beta-lactamases (4.6% and 1%, respectively). Furthermore, we have compared the ESBLs types and associated lineages detected in both groups (CTX-M-15,27,9 in healthy cats, and CTX-M-1,15,55 in sick cats or CMY-2 in both groups). This information is included in abstract, as well as in results and discussion sections (Results: lines 130; 167-168; 176-177, Discussion section: lines 240-243; 303-308). Moreover, the first paragraph of conclusion was deleted, as suggested (411-418). The reference section has been shortened; we have eliminated some of the references and we have added some others suggested by reviewer 2, but the final number is lower than in the initial version (a final reduction of 4 references).

Reviewer 2 Report

Dear Authors,

This manuscript reveals interesting results related to antimicrobial-resistant bacteria in healthy and sick cats, scarcely studied topic, with strong potential enrich this Issue, including in One Health perspective. 

However, I was expecting more data regarding AmpC beta-lactamases and associated mobile genetic elements. Furthermore, while reading the manuscript it seems that the focus is ESBLs epidemiology. 

The focus of this Special Issue s very clear: “occurrence and molecular epidemiology of bacterial species carrying acquired AmpC beta-lactamases within and between different hosts and environments... research papers on the detection and treatment of infections in a clinical setting, whole-genome-based studies on population and mobile genetic elements encoding acquired AmpC beta-lactamases, or risk assessment of antibiotic resistance transmission are also encouraged”.  

One clear example is the objective of this study (Introduction section): ..."It aimed to characterize the antibiotic-resistance profile and the genetic mechanisms in ESBL-producing E. coli isolates..." and AmpC???

This manuscript has a strong potential to be published in this journal but I have doubts regarding in this specific Especial Issue. Thus, if the authors still want to submit this manuscript in this special issue changes are needed. I believe that more data, requiring molecular methods, and information related to AmpC epidemiology, are required in order to strengthen the relevance of acquired AmpC beta-lactamase and the possibility of the transmission to Humans.

Thereby, please address the following specific comments/suggestions:

Introduction section:

In general, I recommend a review of this section, mainly adding more data/information in order to strengthen the relevance of this study.

- The authors talk about the relevance of uropathogenic E. coli. So,add information about STEC E. coli to justify why You did the detection of specific virulence genes associated to this E. colienteropathogenic pathotype.

- Add information about the most prevalent and relevant qAmpCs and associated E. coliclonal linages, as You did for ESBLs.

- Line 83: Please add more recent references and with more data regarding AMR E. coliin Portugal, specially recovered from humans, including qAmpCs. Last clinical human report in Portugal from 2004? I found other 3 from the same Portuguese scientific group of the reference 35, which also includes qAmpCs and/or humans from community.

-Lines 89: Alterations are needed regarding the aim of this study in order to be in line with this specific Special Issue: “occurrence and molecular epidemiology of bacterial species carrying acquired AmpC beta-lactamases within and between different hosts and environments”

(https://www.mdpi.com/journal/antibiotics/special_issues/epidemiology_)

-Figure 1:It is missing the kennel and veterinary clinics/Hospital: B, F, G, H. Furthermore, I think A, C, D, E are not in the correct place. Legend: Replace Braganza to Bragança, as You have in the text and figure.

 -Table 1: You have in the legend the clinic CVV but is missing in the table. Replace Clínica veterinária to Veterinary clinic as You have in the text and figure 1. Try to standardize these types of concepts in all manuscript.

Discussion section:

Lines 220-223: The authors have only 4 lines discussing the detected CMY-2-producing E. coli.The relevance of the clonal linage ST429? Add information.  Once again, the focus of this issue is qAmpCs. Add information regarding its global prevalence. Compare qAmpCs detected in different ecological niches in Portugal, in order to understand and demonstrate the potential transmission to humans. If You had studied the mobile genetic elements associated with the genes, you would have more relevant data demonstrating the circulation of the plasmids in the different ecological niches, which is very important in the ONE HEALTH perspective.

Material and Methods Section:

-I suggest to study the genetic mobile elements associated with the detected beta-lactamase genes. These missing data will clearly enrich this study.

Line 240:You selected only 1 colony per sample. Why only one colony? Then, it is impossible to analyse the potential diversity of beta-lactamase genes in the same sample. With this type of isolates selection  probably You lost other relevant beta-lactamase genes.

Lines 254-256: The screening of phenotypic ESBL production. Why not also study the phenotypic AmpC production?

-Lines 260-263:“Depending on the resistance phenotypes of the isolates, they were screened for the presence of the beta-lactamase genes (blaTEM,blaSHV, blaOXA, blaCTX-M of different groups, blaCMY and blaDHA), as well as for the genes tetA, tetB, aac(3)-II,cmlA, mcr-1, mcr-4 and sul3 [66-69].” You studied resistance genes according to the resistance phenotype obtained, so:

 1-It is missing Tetracycline in the susceptibility testing section.

2- Why you try to detect the variants mcr-1 and mcr-4 genes if You did not study the susceptibility to colistin? It is already reported 10 variants of mcr gene, Why only study the presence of these 2 variants?

3- Why you did not study the presence of PMQRs genes or intrinsic resistance mechanism, mutations in gyrA, gyrA, parC, parE to justify the resistance to Ciprofloxacin?

4- Why only sul3 gene for Sulfamethoxazole? Thesul1gene is more frequently associated in class 1 integrons. Furthermore, according to the table the presence of int1 is associated with resistance to trimethoprim (dfr variant genes), sulfamethoxazole (sul1) and/or streptomycin (aadA genes), typic of class 1 integrons.

5- Regarding chloramphenicol resistance, why only study the presence of cmlA gene,iffloR andcatA genes are frequent too. Or maybe the PMQR genes oqxA-oqxB, all isolates resistant tochloramphenicol present also resistance to fluoroquinolones.

Conclusion Section: Nothing about AmpC? One Health perspective? (The focus of this Issue!)

Acknowledgments: In English Dra. is incorrect, replace for Dr.

Author Response

This manuscript reveals interesting results related to antimicrobial-resistant bacteria in healthy and sick cats, scarcely studied topic, with strong potential enrich this Issue, including in One Health perspective. However, I was expecting more data regarding AmpC beta-lactamases and associated mobile genetic elements. Furthermore, while reading the manuscript it seems that the focus is ESBLs epidemiology. 

The focus of this Special Issue is very clear: “occurrence and molecular epidemiology of bacterial species carrying acquired AmpC beta-lactamases within and between different hosts and environments... research papers on the detection and treatment of infections in a clinical setting, whole-genome-based studies on population and mobile genetic elements encoding acquired AmpC beta-lactamases, or risk assessment of antibiotic resistance transmission are also encouraged”.  

One clear example is the objective of this study (Introduction section): ..."It aimed to characterize the antibiotic-resistance profile and the genetic mechanisms in ESBL-producing E. coli isolates..." and AmpC???

This manuscript has a strong potential to be published in this journal, but I have doubts regarding in this specific Especial Issue. Thus, if the authors still want to submit this manuscript in this special issue changes are needed. I believe that more data, requiring molecular methods, and information related to AmpC epidemiology, are required in order to strengthen the relevance of acquired AmpC beta-lactamase and the possibility of the transmission to Humans.

Thereby, please address the following specific comments/suggestions:

Introduction section:

In general, I recommend a review of this section, mainly adding more data/information in order to strengthen the relevance of this study.

The authors talk about the relevance of uropathogenic E. coli. So,add information about STEC E. coli to justify why You did the detection of specific virulence genes associated to this E. coli enteropathogenic pathotype.

Add information about the most prevalent and relevant qAmpCs and associated E. coli clonal linages, as You did for ESBLs.

Answer: Thank you for the general suggestions and corrections. We have modified the introduction section as suggested, indicating the interest of studying STEC in this type of animals, and also focusing the interest in the qAmpC beta-lactamase detection and epidemiology, in addition to ESBL characterization (see lines 75-98; 106-109). In this study, we have detected ESBL-producing isolates in a higher proportion than qAmpC-producing isolates, but it is relevant the detection of a CMY-2-producing isolates both in healthy and in sick animals, what has been highlighted.

- Line 83: Please add more recent references and with more data regarding AMR E. coli in Portugal, specially recovered from humans, including qAmpCs. Last clinical human report in Portugal from 2004? I found other 3 from the same Portuguese scientific group of the reference 35, which also includes qAmpCs and/or humans from community.

Answer: We have included in the new version of the manuscript four new references related to the detection of qAmpC in E. coli isolates in Portugal, including the three ones suggested by the reviewer. We have included the information in the introduction section (lines 75-98), and we think that this inclusion gave a better picture of the situation in Portugal.

-Lines 89: Alterations are needed regarding the aim of this study in order to be in line with this specific Special Issue: “occurrence and molecular epidemiology of bacterial species carrying acquired AmpC beta-lactamases within and between different hosts and environments”

(https://www.mdpi.com/journal/antibiotics/special_issues/epidemiology_)

Answer: The reviewer is correct. The aim of this study was uploaded to “analyze the occurrence and molecular epidemiology of qAmpC- and ESBL-producing E. coli isolates, originating from healthy cats (kennel and house cats) and sick cats (of seven different hospitals) in Portuguese territory.” (lines 121-126).

-Figure 1: It is missing the kennel and veterinary clinics/Hospital: B, F, G, H. Furthermore, I think A, C, D, E are not in the correct place. Legend: Replace Braganza to Bragança, as You have in the text and figure.

Answer: The reviewer is correct. These data were moved at the moment of submission and they were not in the correct place of Figure 1. The changes were done as recommended in the manuscript (Figure 1 and its legend).

 -Table 1: You have in the legend the clinic CVV but is missing in the table. Replace Clínica veterinária to Veterinary clinic as You have in the text and figure 1. Try to standardize these types of concepts in all manuscript.

Answer: The veterinary clinic names were modified and standardize in the manuscript. Regarding the CVV, it is not present in any elements of the table. For this reason, the concept was not included in the Table 1. The manuscript text was changed (lines 137-140), and the table legend was updated as following:

HC- Healthy cats from their owners; VCQ - Veterinary clinic Quinchosos (Vila Real, Portugal); VCT - Veterinary clinic Transmonvete (Vila Real, Portugal); VCM - Veterinary clinic Macedo de Cavaleiros (Bragança, Portugal); VCG - Veterinary clinic Guia (Leiria, Portugal); VCV - Veterinary clinic Vouga (Sever do Vouga, Portugal).

Discussion section:

Lines 220-223: The authors have only 4 lines discussing the detected CMY-2-producing E. coli. The relevance of the clonal linage ST429? Add information.  Once again, the focus of this issue is qAmpCs. Add information regarding its global prevalence. Compare qAmpCs detected in different ecological niches in Portugal, in order to understand and demonstrate the potential transmission to humans. If You had studied the mobile genetic elements associated with the genes, you would have more relevant data demonstrating the circulation of the plasmids in the different ecological niches, which is very important in the ONE HEALTH perspective.

Answer: Thank you for the suggestion. General information related with AmpC was completed in the manuscript, as well as data in Portugal was updated. Particularly, recent researches with detection of CMY-2 gene among Portuguese clinical and non-clinical samples were discussed (lines 215; 226-228). Furthermore, new information was added regarding the relevance of the clonal lineage ST429 (lines 305; 319-332).

Material and Methods Section:

Line 240: You selected only 1 colony per sample. Why only one colony? Then, it is impossible to analyse the potential diversity of beta-lactamase genes in the same sample. With this type of isolates selection  probably you lost other relevant beta-lactamase genes.

Answer: Reviewer is right that perhaps we could have lost some isolates with other ESBL or qAmpC beta-lactamases of relevance, and this could be a limitation of the study; we have included this point in the discussion section (lines 210-212). Our purpose was to determine the rate of samples with ESBL or qAmpC-producing E. coli isolates, as well as the more predominant beta-lactamases among faecal samples of cats, knowing that with this strategy we could have lost some ESBL or pAmpC variants (probably in minority, if present). 

Lines 254-256: The screening of phenotypic ESBL production. Why not also study the phenotypic AmpC production?

Answer: We have clarified the phenotypic strategy that we followed to detect presumptive qAmpC among selected CTXRisolates, that later on they were tested for the presence of qAmpC encoding genes in order to confirm the qAmpC-producers. In the initial study, we analyzed the presence of qAmpC not only among the ESBL-negative isolates but now we have tested among all the FOX-R isolates (including ESBL-producers) and we have detected two new CMY-2 positive isolates that have been included in the final version of the manuscript. Taking together, the prevalence of qAmpC among the total samples of cats analyzed is 2.1% (4.6% in healthy animals and 1% in sick animals). These new data are included in the revised version of the manuscript (lines 174-179; 205-206).

-Lines 260-263:“Depending on the resistance phenotypes of the isolates, they were screened for the presence of the beta-lactamase genes (blaTEM, blaSHV, blaOXA, blaCTX-M of different groups, blaCMY and blaDHA), as well as for the genes tetA, tetB, aac(3)-II,cmlA, mcr-1, mcr-4 and sul3 [66-69].” You studied resistance genes according to the resistance phenotype obtained, so:

 1-It is missing Tetracycline in the susceptibility testing section.

Answer: The reviewer is correct. It was an error not to include it. We have included it in Material and Methods section (line 364).

2- Why you try to detect the variants mcr-1 and mcr-4 genes if You did not study the susceptibility to colistin? It is already reported 10 variants of mcr gene, Why only study the presence of these 2 variants?

Answer: According with new guidelines from CLSI (2019), the disk diffusion test for colistin is not recommended to evaluate susceptibility or resistance to this antimicrobial drug. So, we did not test colistin by disk-diffusion, but we tried to determine if the isolates carried any of the most frequent mcr variants in E. coli (specially mcr-1, but also mcr-4).

3- Why you did not study the presence of PMQRs genes or intrinsic resistance mechanism, mutations in gyrAgyrA, parC, parE to justify the resistance to Ciprofloxacin?

Answer: We have focused to mechanisms that are transferable and for this reason we have not analyzed the mutations in intrinsic genes. In relation with ciprofloxacin, we have analyzed the presence of the aac(6´)-Ib-cr among ciprofloxacin-resistant isolates, and two of them carried this gene. This information is now included in the revised version of the manuscript (lines 185-186; 390).

4- Why only sul3 gene for Sulfamethoxazole? The sul1gene is more frequently associated in class 1 integrons. Furthermore, according to the table the presence of int1 is associated with resistance to trimethoprim (dfr variant genes), sulfamethoxazole (sul1) and/or streptomycin (aadA genes), typic of class 1 integrons.

Answer: The reviewer is correct. We have tested the presence of sul1 and sul2 genes among the trimethoprim/sulfamethoxazole-resistant isolates. These genes were also added in Material and Methods section (line 389). One trimethoprim+sulfamethoxazole-resistant isolate was positive for sul1, another isolate for sul2, and the combination of both sul1/2 was detected among 2 additional isolates. We could not go further in the characterization of integrons. This new research data was included in the Results section (lines 185-186), as well as in Table 1.

5- Regarding chloramphenicol resistance, why only study the presence of cmlA gene,if floR  and catA genes are frequent too. Or maybe the PMQR genes oqxA-oqxB, all isolates resistant to chloramphenicol present also resistance to fluoroquinolones.

Answer: According with suggestion of reviewer, we have tested the presence of cat and florR genes among chloramphenicol-resistant isolates and all were negative. In addition, the aac(6´)-Ib-cr gene was analyzed among ciprofloxacin-resistant isolates and two isolates were positive. These new data were added in Material and Methods section (lines 387-391).

Conclusion Section: Nothing about AmpC? One Health perspective? (The focus of this Issue!)

Answer: The reviewer is correct. Information related qAmpC and One health perspective was updated conclusions section (lines 422-433).

Acknowledgments: In English Dra. is incorrect, replace for Dr.

Answer: The change was done in the manuscript (lines 443-444).

Reviewer 3 Report

General comments

This study analyzes the mechanisms of resistance in ESBL, and AmpC-producing E. coli isolates from healthy and sick cats in Portugal. The authors use a combination of traditional phenotypic and PCR/sequencing methods to characterize these isolates. They compared their results according to previous literature and analyzed them in this context. The authors claim that this is the first report combining healthy and sick cats in Portugal and that their findings represent a public health problem due to the proximity of cats and humans

Major concerns

¿Why did the authors restricted their study to ESBL and AmpC? By only cultivating their samples on MacConkey agar supplemented with cefotaxime, they lost the opportunity to find many other resistance mechanisms relevant to public health, including many mobile resistant determinants. From 141 samples, only 13 isolates were finally studied. No information was obtained from the other 128 samples.

The authors indicate that this is the first report combining healthy and sick cats in Portugal. Why did they consider it relevant to combine healthy and sick animals in the same study? What was their hypothesis? The isolates obtained from sick or healthy animals were not analyzed separately. The authors do not offer a significant conclusion based on this particular variable.

Minor concerns

What is the relevance of Figure 1. Besides knowing the samples' location, this information is not analyzed or discussed further in the paper.

The data should be analyzed based on the latest CLSI guide, not 2018.

Author Response

General comments

This study analyzes the mechanisms of resistance in ESBL, and AmpC-producing E. coli isolates from healthy and sick cats in Portugal. The authors use a combination of traditional phenotypic and PCR/sequencing methods to characterize these isolates. They compared their results according to previous literature and analyzed them in this context. The authors claim that this is the first report combining healthy and sick cats in Portugal and that their findings represent a public health problem due to the proximity of cats and humans

 Major concerns

¿Why did the authors restricted their study to ESBL and AmpC? By only cultivating their samples on MacConkey agar supplemented with cefotaxime, they lost the opportunity to find many other resistance mechanisms relevant to public health, including many mobile resistant determinants. From 141 samples, only 13 isolates were finally studied. No information was obtained from the other 128 samples.

The authors indicate that this is the first report combining healthy and sick cats in Portugal. Why did they consider it relevant to combine healthy and sick animals in the same study? What was their hypothesis? The isolates obtained from sick or healthy animals were not analyzed separately. The authors do not offer a significant conclusion based on this particular variable.

 Minor concerns

What is the relevance of Figure 1. Besides knowing the samples' location, this information is not analyzed or discussed further in the paper.

The data should be analyzed based on the latest CLSI guide, not 2018.

Answer: Thank you for your comments. The purpose of our study was to analyze the isolates showing resistance to third generation cephalosporins (using CTXR as a marker) and for this reason we used this strategy. We know that we could have used a wider strategy looking for other resistance markers, but we restricted to CTXR isolates in this study. In any case, we consider that is of relevance the detection and characterization of ESBL- and qAmpC-producers in the population of cats, that are very close to humans and that animal-human transference of resistance bacteria could occur. In this sense, we analyzed the 13 CTXR isolates that were grown in MacConkey plates supplemented with cefotaxime, but the remaining faecal samples for which there was no growth, they were not further studied.

On the other way, we were interested in comparing in the same study and with the same methodology, both sick and healthy cats. In the revised version of the manuscript, we have presented in a more clear and extended way the comparative data of healthy and sick animals (results and discussion section and tables: lines 130; 167-168; 176-177; 240-243; 303-308). Moreover, CLSI (2019) data was updated.

Round 2

Reviewer 1 Report

The authors modified the whole manuscript according to the reviewers's comments.

Author Response

Thank you for your comments.

Reviewer 2 Report

Dear Authors,

After reading this new version of the manuscript, I concluded that the authors answered well to my questions and considered my suggestions.  After alterations, now I think that the current version of this manuscript is in line with this specific Special Issue. For this reason, I recognize and congratulate the authors efforts to restructure this manuscript.

Clarifying some answers:

1- “According with new guidelines from CLSI (2019), the disk diffusion test for colistin is not recommended to evaluate susceptibility or resistance to this antimicrobial drug. So, we did not test colistin by disk-diffusion, but we tried to determine if the isolates carried any of the most frequent mcr variants in E. coli (specially mcr-1, but also mcr-4).”

I understood the reason why You only study the presence of mcr-1 and mcr-4. However, despite the disk diffusion test is not recommended to evaluate the susceptibility to colistin. You could perform the ISO-standard broth microdilution method (20776-1), as recommended by CLSI-EUCAST joint Polymyxin Breakpoints Working Goup since 2016.

2-“According with suggestion of reviewer, we have tested the presence of cat and florR genes among chloramphenicol-resistant isolates and all were negative. In addition, the aac(6´)-Ib-cr gene was analyzed among ciprofloxacin-resistant isolates and two isolates were positive. These new data were added in Material and Methods section (lines 387-391).”

The presence of the gene the aac(6´)-Ib-cr is commonly detected co-present with other PMQR gene. In the 1st revision I referred oqxAB gene, which is usually detected associated with aac(6´)-Ib-cr, because it encodes for an efflux pump that confers susceptibility decrease or resistance not only to fluoroquinolones but also to other antibiotics, as chloramphenicol an tetracycline. That’s why I suggested not only the detection of catA and floR but also oqxAB.

I will not impose the performance of these analysis. Nevertheless, I recommend the authors to read again the manuscript in order to detect any mistakes.

Corrections:

-Keywords, lines 68 and 151: "aAmpC" correct to qAmpC.

-Line 168:"cat" correct to catA.

-Line 320: “resistance phenotypes,…” correct to Resistance phenotypes.

Author Response

1- “According with new guidelines from CLSI (2019), the disk diffusion test for colistin is not recommended to evaluate susceptibility or resistance to this antimicrobial drug. So, we did not test colistin by disk-diffusion, but we tried to determine if the isolates carried any of the most frequent mcr variants in E. coli (specially mcr-1, but also mcr-4).”

I understood the reason why You only study the presence of mcr-1 and mcr-4. However, despite the disk diffusion test is not recommended to evaluate the susceptibility to colistin. You could perform the ISO-standard broth microdilution method (20776-1), as recommended by CLSI-EUCAST joint Polymyxin Breakpoints Working Goup since 2016.

Answer: Thank you for your comments and suggestions. In the present moment it is complicated to do the MIC microdilution for colistin, nevertheless, we plan to analyze colistin susceptibility of a wide collection of E. coli isolates of different origins and we will include those of cat origin. 

2-“According with suggestion of reviewer, we have tested the presence of cat and florR genes among chloramphenicol-resistant isolates and all were negative. In addition, the aac(6´)-Ib-cr gene was analyzed among ciprofloxacin-resistant isolates and two isolates were positive. These new data were added in Material and Methods section (lines 387-391).”

The presence of the gene the aac(6´)-Ib-cr is commonly detected co-present with other PMQR gene. In the 1st revision I referred oqxAB gene, which is usually detected associated with aac(6´)-Ib-cr, because it encodes for an efflux pump that confers susceptibility decrease or resistance not only to fluoroquinolones but also to other antibiotics, as chloramphenicol an tetracycline. That’s why I suggested not only the detection of catA and floR but also oqxAB.

I will not impose the performance of these analysis. Nevertheless, I recommend the authors to read again the manuscript in order to detect any mistakes.

Answer: The reviewer is correct, and we would like to test the oqxAB gene. Unfortunately, it was not possible because of the actual pandemic situation. We did not have the primers available to test this gene, the transport is really slow actually and times cannot be guaranteed. So, we performed catA and floR (after first revision) but it is not possible to test oqxAB gene on time.

Corrections:

-Keywords, lines 68 and 151: "aAmpC" correct to qAmpC.

-Line 168:"cat" correct to catA.

-Line 320: “resistance phenotypes,…” correct to Resistance phenotypes.

Answer: The minor corrections suggested by reviewer 2 were updated in the manuscript (marked with green color).